# Effect Sizes, Power, and Biases in Intelligence Research: A Meta-Meta-Analysis

**DOI:** 10.3390/jintelligence8040036

**Published:** 2020-10-02

**Authors:** Michèle B. Nuijten, Marcel A. L. M. van Assen, Hilde E. M. Augusteijn, Elise A. V. Crompvoets, Jelte M. Wicherts

**Affiliations:** 1Department of Methodology & Statistics, Tilburg School of Social and Behavioral Sciences, Tilburg University, Warandelaan 2, 5037 AB Tilburg, The Netherlands; m.a.l.m.vanassen@tilburguniversity.edu (M.A.L.M.v.A.); h.e.m.augusteijn@tilburguniversity.edu (H.E.M.A.); e.a.v.crompvoets@tilburguniversity.edu (E.A.V.C.); j.m.wicherts@uvt.nl (J.M.W.); 2Section Sociology, Faculty of Social and Behavioral Sciences, Utrecht University, Heidelberglaan 1, 3584 CS Utrecht, The Netherlands

**Keywords:** meta-meta-analysis, meta-science, bias, intelligence, power, effect size

## Abstract

In this meta-study, we analyzed 2442 effect sizes from 131 meta-analyses in intelligence research, published from 1984 to 2014, to estimate the average effect size, median power, and evidence for bias. We found that the average effect size in intelligence research was a Pearson’s correlation of 0.26, and the median sample size was 60. Furthermore, across primary studies, we found a median power of 11.9% to detect a small effect, 54.5% to detect a medium effect, and 93.9% to detect a large effect. We documented differences in average effect size and median estimated power between different types of intelligence studies (correlational studies, studies of group differences, experiments, toxicology, and behavior genetics). On average, across all meta-analyses (but not in every meta-analysis), we found evidence for small-study effects, potentially indicating publication bias and overestimated effects. We found no differences in small-study effects between different study types. We also found no convincing evidence for the decline effect, US effect, or citation bias across meta-analyses. We concluded that intelligence research does show signs of low power and publication bias, but that these problems seem less severe than in many other scientific fields.

## 1. Introduction

Mounting evidence suggests that the literature in psychology and related fields paints an overly positive picture of effects and associations. Many published findings could not be replicated in novel samples ([25]; [41]; [56]; [85]; [115]), many meta-analyses highlight selective reporting of results depending on significance ([14]; [31]; [76], [77]), and the number of confirmed hypotheses in the literature is incompatible with the generally small sample sizes in psychological studies ([8]; [28]; [34]; [66]). It is argued that the main cause for this “replicability crisis” ([5]) is a combination of publication bias and strategic use of flexibility in data analysis ([46]; [75]). Publication bias is the phenomenon where statistically significant results have a higher probability of being published than non-significant results ([40]). Concerning flexibility in data analysis, it is suspected that many researchers try out multiple analysis strategies to search for a significant finding, and only report the ones that “worked” ([8]; [54]; [98]). This increases false positive rates and generally inflates estimates of genuine effects. Because both biases might negatively affect the trustworthiness of published findings, it is important to assess their severity in different bodies of literature. In this paper, we investigated effect size, power, and patterns of bias in the field of intelligence research.

Intelligence research provides a good field to study effect size, power, and biases because it encompasses a wide range of study types using different methods that still focus on measures of the same construct. Intelligence is among the most well-known constructs in psychology and has been investigated extensively from various angles since the development of the first successful intelligence tests in the early 20th century ([10]; [43]; [64], for reviews, see, e.g., [93]). Individual differences in intelligence and cognitive ability tests have been related to many relevant outcomes, correlates, and (potential) causes in the contexts of education, health, cognitive development and aging, economic outcomes, genes, and toxic substances (e.g., adverse effects of lead or alcohol exposure). Intelligence research is a multidisciplinary field with links to behavior genetics, educational sciences, economics, cognitive psychology, neuroscience, and developmental psychology. These different types of research use different methods and involve different effect sizes, and hence might differ in how strongly they are affected by potential biases ([46]). For instance, effect sizes are expected to be fairly large in research relating one type of cognitive test (e.g., fluid reasoning tasks) to another type of cognitive test (e.g., spatial ability tasks) because of the well-established phenomenon of the positive manifold (e.g., [110]). Conversely, research that attempts to improve intelligence by certain interventions might show smaller effects in light of longstanding challenges in raising intelligence (e.g., [100]). Similarly, some research methods in the study of intelligence are more challenging in terms of data collection (e.g., neuroscientific measures, twin designs in behavior genetics, or controlled interventions) than other research methods (e.g., studies that establish correlations between existing measures in readily accessible samples). This may create variations in sample sizes that play a key role in power and (over)estimation of effects and associations.

One way to investigate bias in science is by analyzing patterns in effect size estimates in meta-analyses (see, e.g., [29]; [31]; [52]; [107]). A relevant example is the recent meta-meta-analysis by [91] ([91]), in which they analyzed patterns of bias in 36 meta-analyses (comprising 1391 primary effect sizes) published in the journal *Intelligence*. They found that a majority of individual meta-analyses contained evidence for declining effect sizes over time but no significant, overall decline effect. Our work complements the work of [91] ([91]) by analyzing a larger and more diverse sample of meta-analyses in intelligence research and by analyzing more patterns of bias. Specifically, we analyzed 2442 primary effect sizes from 131 meta-analyses in intelligence research, published between 1984 and 2014, to estimate the average effect size, median power, and evidence of small-study effects, decline effects, US effects, and citation bias. Our data, analysis scripts, and an online Appendix with supplemental analyses and results are available from https://osf.io/z8emy/.

## 2. Method

### 2.1. Sample

We searched for meta-analyses about IQ and intelligence on ISI Web of Knowledge (one of the most well-known interdisciplinary databases) on 29 August 2014. Using the search string “TS = (IQ OR intelligence) AND TS = (meta-analysis)” we found 638 records. From these 638 records, we excluded six duplicate records and 21 records that were not available in English. We then looked at the abstracts of the articles and excluded a further 208 articles that did not report a quantitative meta-analysis, and we excluded another 107 meta-analyses that did not use IQ tests or other cognitive maximum performance tests that were featured in Carroll’s (1993) seminal review of the intelligence literature ([16]). Finally, we excluded 28 records that were only available in print format or put behind paywalls that our library had no subscription to. The articles we could not access were diverse in terms of publication year, impact factor, and study type, which leads us to suspect that these excluded articles are not systematically different from the included ones. Note also that not all meta-analyses eventually offered useful data. Therefore, we do not suspect that adding the handful of meta-analyses from these excluded articles would have affected our conclusions substantially. In the end, we obtained a set of meta-analyses spanning 30 years, with primary studies published from 1915 to 2013, spanning almost a century.

All effect sizes retrieved from the meta-analyses were based on independent samples both within and between meta-analyses (below we indicate how we dealt with potential overlap between meta-analyses). Some meta-analyses reported the results from different types of cognitive tests from the same sample. If the meta-analysis reported results for Full Scale IQs (FSIQs), we included only those FSIQs. If the meta-analysis reported only Verbal IQs or Performance IQs, we selected one of these, depending on which set had the largest number of primary studies. We do note that effect sizes based on subtests rather than FSIQs will be noisier. If no IQ measure was presented in the meta-analysis, we chose the largest set that used a cognitive test (or a set of similar cognitive tests) that was strongly associated with the general factor of intelligence ([71]). Thus, whenever the meta-analysis lacked IQ measures, we included studies that used the same cognitive test (e.g., the use of Raven’s Progressive Matrices Test) or highly similar tests that were labeled in the same manner in the meta-analytic article (e.g., all used fluid reasoning tasks). Because of the positive manifold between cognitive tests and fairly high correlations between broad cognitive factors ([71]), this strategy ensured inclusion of measures bearing on intelligence, while also creating less heterogeneity within meta-analyses than would have been present had we included entirely different (types of) cognitive tests.

One article contained two independent meta-analyses, so we included both. We excluded 102 meta-analyses because they did not contain sufficient data (or any data at all) to calculate the effect sizes and standard errors for the primary studies and 32 meta-analyses that used non-standard meta-analytic methods (e.g., multilevel models based on individual level data or unweighted analyses) for which we could not calculate primary effect sizes that were suitable for this meta-meta-analysis. Finally, we excluded nine meta-analyses that contained fewer than three unique primary studies, because it was not possible to estimate patterns of bias based on only two observations. The resulting sample consisted of 131 meta-analyses consisting of 2446 unique primary studies. For four primary studies, we were not able to calculate the effect size, so these were excluded from our analyses. Our final sample consisted of 2442 primary studies, with over 20 million participants in total. Figure 1 shows a schematic overview of the exclusion criteria and meta-analysis selection. A list of all included meta-analyses can be found at the end of the paper.

### 2.2. Procedure

**Variables:** For each meta-analysis, we coded several variables. First, we coded whether primary studies were unique in our sample, to avoid dependencies between meta-analyses. If a study appeared in more than one meta-analysis, we removed it from the meta-analysis with the most primary studies. This way, we ensured that the number of effect sizes of the individual meta-analyses would remain as large as possible.

Furthermore, for each unique primary study, we recorded the effect size that was included in the meta-analysis and its standard error (SE). Per primary study, we only included one effect size where we prioritized effects based on FSIQ, VIQ, or PIQ, or other cognitive tests, in that order (see above for details). Often, the meta-analysts calculated the effect size and its SE of a primary study themselves. Analyzing data and reporting results are error prone (see, e.g., [7]; [39]; [63]; [67]; [82]; [88]). To minimize the risk of copying erroneously calculated or reported effect sizes and SEs, we calculated the effect sizes and SEs ourselves using data reported in the meta-analysis, where possible. In these cases, we did not record the effect size and SE reported by the authors. It would be an interesting additional study to estimate how much reported and recalculated effect sizes and SEs differed, but this was beyond the scope of this study.

Effect sizes could often be calculated with statistics such as means and standard deviations or frequency tables, and we could often calculate the SE using sample sizes or confidence intervals. If there was insufficient information available to calculate the primary studies’ effect size and SE, we copied them directly from the meta-analysis. Where possible, we also recorded the primary studies’ total sample size and the sample size per condition. After a first round of data collection, all effect size computations and effect size retrievals from meta-analytic articles were checked by a second coder to avoid errors and to correct any errors that emerged.

Finally, we categorized the meta-analyses in five different types of research: correlational, group differences, experiments and interventions, toxicology, and behavior genetics (see Table 1). We chose these five categories to distinguish between substantively different types of research questions and their associated research designs, while retaining a sufficient number of meta-analyses in each type to make reliable inferences about effect sizes, power, and bias.

Correlational studies refer to studies that lack any manipulation or treatment, in which a measure of intelligence is correlated with another individual difference variable that is measured on a continuous scale. The effect sizes in such studies are typically Pearson’s correlations. Examples of such studies include studies relating IQ to personality ([21]), brain size ([70]), or self-rated intelligence ([36]). Studies on group differences compare existing (non-manipulated) groups and typically use Cohen’s *d* or raw mean IQ differences as the key effect size. Examples include studies comparing mean IQs between men and women ([50]) or mean IQs between healthy controls and people diagnosed with schizophrenia ([4]). Experiments and interventions are studies that attempt to improve the IQ of either healthy or unhealthy groups. Effect sizes are typically standardized mean differences, and examples include studies investigating the effect of interventions improving cognitive development in institutionalized children ([6]), or the effect of iron supplementation on cognition ([27]). Studies of toxic effects on IQ entail observational studies or randomized clinical trials, in which the toxic effects relate to side effects of a certain drug treatment, often expressed in standardized mean differences. Examples include studies investigating potential harmful effects of lead exposure on IQ ([15]) or prenatal cocaine exposure on children’s later IQ ([61]). Finally, behavior genetic studies link intelligence to genetic variations or estimate heritability using twin designs. Effect sizes are often expressed in log odds ratios or Pearson’s *r*. Examples include studies about the heritability of cognitive abilities ([9]) or studies linking specific genetic variants to general cognitive ability ([118]).

We ordered the types in increasing complexity of the methodology. Correlational studies and studies about group differences usually do not require special populations and often make use of convenience samples. In experimental research, the methodology increases in complexity because participants have to be randomly assigned to carefully constructed conditions. Toxicological studies are mainly quasi-experimental but require special populations, which makes them logistically much more challenging. Finally, behavior genetic studies are arguably the most complex in methodology and often require special groups (especially in twin designs). The five study types were independently coded by MN and JW. The initial interrater reliability was a Cohen’s κ = 0.868. Any coding discrepancies were solved through discussion by the coders.

The study categorization was not defined a priori but rather decided upon ad hoc. It is possible to come up with reasonable alternative categorizations. For instance, another option would be to classify studies purely according to research design: correlational, quasi-experimental, and experimental. Effectively, this would mean combining studies on group differences and toxicological studies into one category (quasi-experimental) and distributing the meta-analyses in behavior genetics over correlational and quasi-experimental studies. This alternative categorization did not substantively affect the median sample sizes and effect sizes (see Appendix A). However, we would like to emphasize that our categorization is one of many possibilities, and other categorizations may show different patterns in effect size, power, and bias. Our open data file allows readers to pursue other categorizations and analyses. Finally, we excluded eight primary studies that occurred in multiple meta-analyses of different types, but, after rerunning some of our descriptive analyses, we concluded that this is unlikely to have affected our conclusions (see Appendix A).

**Effect size conversion:** For our analyses, we converted the effect sizes in all meta-analyses to a single type of effect size. For most of the meta-analyses, the effect size we extracted or calculated based on available data was either a Cohen’s *d* (80 meta-analyses; 61.1%) or a correlation (*r*; 42 meta-analyses; 32.1%), so converting to one of these effect sizes seemed most convenient. We chose to convert all effect sizes to *r* because it made more conceptual sense to express a *d* in *r* than vice versa. If one expresses a *d* in *r*, the resulting point biserial correlation gives exactly the same information as *d*, but if one expresses an *r* in *d*, the *d* loses information (for more information, see [94]). For the meta-analyses of patterns of bias, we subsequently converted all *r* values to Fisher’s *Z* values, because the standard error then only depended on the sample size and not on the correlation itself (see also [103]).

The direction in which the meta-analytical hypothesis was formulated can affect whether the primary effect sizes are reported as positive or negative. To correct for any influence of the direction of the hypothesis, we used a procedure called “coining,” following [31] ([31]). In this procedure, we checked all meta-analytic articles that yielded a negative average meta-analytic effect and concluded that in all of these cases the result was in line with the expectations of the meta-analysts. In these cases, we multiplied all primary effect sizes within the meta-analysis by −1. After recoding, all primary studies with negative outcomes showed an effect contradicting the hypothesis of the meta-analysts. The following analyses all used coined primary effect sizes unless stated otherwise. All our data and analysis scripts are freely available from https://osf.io/z8emy/.

## 3. Effect Sizes in Intelligence Research

We were able to convert the effect sizes from a total of 2442 primary studies to Fisher’s *Z* values. For four primary studies, we were not able to convert the effect sizes because information on sample sizes was missing. Figure 2 shows the distribution of the 2442 primary effect sizes converted back to Pearson’s correlations to facilitate interpretation. The unweighted mean effect size in the 2442 primary studies was a Pearson’s correlation of 0.25 (SD = 0.23; slightly higher than the median estimate of 0.19 as observed in individual differences research; [37]), with a minimum of −0.94, and a maximum of 0.95 (i.e., almost the whole range of possible values). The sample size in the primary studies varied widely, from six participants to over 1,530,000, with a median total sample size per primary study of 60. Histograms of the correlations and total sample sizes split up per study type can be found in Appendix A.

We also looked at the sample sizes and effect sizes of the five study types separately and found some clear differences between them (see Table 2). First, the majority of meta-analyses and primary studies concerned either research about group differences in intelligence (59 meta-analyses, 1247 primary studies) or correlational research (31 meta-analyses, 779 primary studies) in which intelligence was related to other continuous psychological constructs. However, we also noted that some meta-analyses seemed to overlap substantially. For instance, in our sample we included 12 meta-analyses about the cognitive abilities in schizophrenia patients. This could be a sign of redundancy in the meta-analyses that are produced in this field, as has been found in medicine research ([48]).

Interestingly, in all different study types we found relatively low median sample sizes, considering the average observed effect sizes per type. This suggested that intelligence research might be generally underpowered. Note, however, that median sample sizes also varied considerably across the study types, with those of behavioral genetics (N = 169) being much larger than for the other four types (N = 49–65). The meta-analytic effect size also differed across the five types. We will come back to this in the next section where we estimate the power across all meta-analyses and for the different study types separately.

## 4. Power in Intelligence Research

Several studies found that sample sizes in psychology research are typically too small to detect the expected true effect sizes (which are also estimated to be small to medium), leading to the conclusion that many psychology studies may be underpowered ([14]; [20]; [97]). Despite repeated recommendations to change this, there seems to have been no overall improvement ([33]; [42]; [66]; [68]; [101]; [105]; [65]; [95]; [107]).

Small sample sizes and low statistical power lead to several problems. First, if a study is underpowered, the chance that it detects a true effect decreases. Second, in a set of studies containing both null effects and genuine effects, a small sample size increases the chance that a significant study represents a false positive ([46]). Third, when a significant finding in an underpowered study does reflect a true effect, it is likely to be overestimated ([14]). These problems occur even when all other research practices are ideal, and there is strong evidence that they are not. Researchers have a strong focus on reporting significant results ([35]; [60]). To obtain significant results they seem to make strategic use of flexibility in data analysis, also referred to as “researcher degrees of freedom” ([1]; [54]; but see [32]; [98]; [117]). Small studies are particularly vulnerable to such researcher degrees of freedom, both because they probably will not find a significant effect in the first place, but also because effect sizes are particularly strongly affected by researcher degrees of freedom in a small-study ([8]).

Here, we estimated the power of the primary studies to detect a small, medium, or large effect. We retained traditional cut-offs in which small, medium, and large effects corresponded to a Pearson’s *r* of 0.1, 0.3, and 0.5 or to a Cohen’s *d* of 0.2, 0.5, and 0.8, respectively. We calculated power using either a *t*-test for correlation (if the primary study’s effect size was a correlation) or a two-sample *t*-test (if the primary study’s effect size was a Cohen’s *d*)[note 1], assuming *α* = 0.05 and two-sided tests, using the R package “pwr” ([18]). Finally, we summarized the power estimates for all primary studies by taking their median.

Overall, we found a median power of 11.9% to detect a small effect, 54.5% to detect a medium effect, and 93.9% to detect a large effect. The power estimates differed substantially between study types (see Table 3). For example, the median power was systematically the lowest for experimental studies: 10.5%, 39.9%, and 77.9% power to detect small, medium, and large effects, respectively. Conversely, studies in behavior genetics seemed to have the highest power of the different study types to detect small, medium, and large effects. However, it is important to note that more than half of the behavior genetics studies reported an effect smaller than what we considered a “small” effect here. This could mean that the actual power of these studies was much lower than the median power we report in Table 3.

In order to try and approach the “real” power of each study, we also estimated the power of each primary study to detect the corresponding meta-analytic effect as a proxy for the “true” effect. Following this strategy, we found an overall median power of 51.7%. Overall, less than one third (30.7%) of all primary studies included in our sample reached the recommended power of 80% or higher. It is important to note that these power estimates were likely biased due to heterogeneity in the true effect sizes and noisy because of the often small number of studies included in each meta-analysis ([55]; [72]; [73]). We report details on this analysis and its results in Appendix A.

It has been argued that estimates of the average power across an entire field lack nuance ([78]) and could paint the possibly misleading picture that *all* studies in a field are underpowered (or adequately powered) to detect a certain effect, which is not necessarily true. Indeed, across the 2442 primary studies, we found that the power of individual primary studies varied widely (see Figure 3).

Figure 3 shows the decumulative proportion of primary studies that reached a certain power or more to detect a small, medium, or large effect, split up per study type and overall. The vertical dashed lines indicate a power of 80% or more. Consider, for example, the top purple line in the first panel in which the overall results are shown. The results show that only 66% of all included primary studies reached a power of 80% or more to detect a large effect (the purple line crosses the vertical line at a cumulative proportion of 0.66). The turquoise and yellow lines in the same panel show that 30% of the studies had sufficient power to detect a medium effect, and only 4% of the studies had sufficient power to detect a small effect.

Figure 3 also shows the distributions of power to detect different effects. For example, compare the yellow lines in the panels for group differences and behavior genetics studies. From the figure, it shows that the bulk of the studies on group differences had very low power (<25%) to detect a small effect, represented by the drop in the curve from 1 to about 0.15, indicating that about 85% of primary studies had very low power. Conversely, in behavior genetics studies, the bend in the curve is less pronounced, with a corresponding drop from 1 to 0.45, indicating that about 55% of primary studies had very low power (<25%) to detect a small effect.

Based on these results we concluded that sample sizes in intelligence research generally seem to be too low to reliably detect hypothetical medium (or small) effects. Based on the observed effect sizes in the primary studies, it is unlikely that many effects in the intelligence literature are larger than medium. As we discussed above, small studies with low power are more at risk to overestimate effect sizes when biases are present (e.g., publication bias, or researcher degrees of freedom; [8]; [81]). It is important to note that simply combining studies in meta-analyses is not sufficient to eliminate any bias from the meta-analytic estimate ([57]; [58]; [81]). If effects in primary studies are overestimated, combining them in a meta-analysis will increase precision (i.e., a smaller standard error) but result in an overall biased estimate. In the next section, we investigate whether different biases are likely to have affected effect size estimates in different types of intelligence research and in intelligence research as a whole.

## 5. Bias-Related Patterns in Effect Sizes

We investigated whether different bias-related patterns were present in intelligence research, starting with the small-study effect. A small-study effect occurs when (published) studies with smaller sample sizes yield larger average effect sizes than those with larger sample sizes ([104]). One possible cause of a small-study effect is publication bias. Smaller studies generally contain more sampling error, which means that the effect size estimates can vary widely. Effects in smaller studies need to be larger than effects in larger studies in order to reach significance thresholds. If mainly the statistically significant effects are published, small studies with overestimated effects will be overrepresented in the literature. In a meta-analysis, such a small-study effect is readily visible by verifying whether the effects in primary studies can be predicted by the studies’ precision.

It is important to note that a small-study effect does not necessarily signify bias. For instance, a small-study effect can also arise because of true heterogeneity in which underlying effects happen to be related to studies’ precision. For instance, in a clinical meta-analysis, the study size may be related to the intensity of the intervention: It may be the case that more strongly afflicted patients are both rare and receive more extensive treatments with larger effects than less afflicted patients who are more numerous. A small-study effect can also arise purely by chance. For an overview of alternative explanations of the small-study effect, see [103] ([104]).

### 5.1. Two-Step Meta-Regression

Small-study effects can be analyzed with two-step meta-regressions ([30]; [31]; [80]). Here, bias-related patterns are investigated for each individual meta-analysis, and this information is then combined across all meta-analyses. We used this two-step strategy here.[note 2]

Within each individual meta-analysis, we estimated the small-study effect with the following meta-regression:(1)$$Fisher^{\prime }s\;Z_{ij}=\;a^{j}+\;b^{j}SE_{ij}+\;\varepsilon_{ij},$$
where the dependent variable *Fisher’s Z_ij_* is the coined effect size of primary study *i* in meta-analysis *j* weighted by its standard error, *a^j^* is the intercept, *SE_ij_* is standard error for the primary study’s effect size, and *b^j^* indicates the unstandardized regression coefficient of *SE_ij_* in predicting *Fisher’s Z*. A positive *b^j^* coefficient would indicate that larger SEs are associated with larger effects, signaling a small-study effect. All meta-regressions were estimated in R ([92]; version 3.6.1) using the rma() function in the R package metafor ([113]). We assumed random effects models, and we used the Paule–Mandel estimator for random effects because it had the most favorable properties in most situations to estimate variance in true effect size between studies ([59]; [111]).

After running this meta-regression for each of the meta-analyses, we obtained estimates of the small-study effects (and their SEs) in the separate intelligence meta-analyses. To combine this information across meta-analyses, we then ran another meta-analysis to obtain a weighted average of all obtained regression coefficients *b^j^*. At this meta-meta-level, we again used the Paule–Mandel estimator for random effects. We assumed random effects models at both levels, because it was highly unlikely that the same population effect underlay (1) every study within the meta-analyses, and (2) every meta-analysis’ small-study effect in the meta-meta regression ([11]).[note 3]

### 5.2. Results of Small-Study Effect

We excluded one meta-analysis from this analysis because there was too little variation in the standard errors of the primary studies to estimate a small-study effect. Across the remaining 130 meta-analyses, we found a significant overall small-study effect, b_SE_ = 0.67, SE = 0.12, *Z* = 5.46, *p* < 0.001, 99% CI = [0.35; 0.99], *I*^2^ = 47.3%, *var* (representing variation in small-study effects across meta-analyses) = 0.73 (SE = 0.25). Concretely, the estimated overall small-study effect of b_SE_ = 0.67 across meta-analyses meant that two otherwise identical studies with sample sizes of 25 and 100 observations would estimate Pearson’s *r* equal to 0.3 (SE = 0.2) and 0.2 (SE = 0.1), respectively.

Even though we found a significant overall small-study effect across meta-analyses, this did not mean that every meta-analysis showed a small-study effect. In 17 of the individual meta-analyses (13.1%) we found a significant small-study effect (*α* = 0.05). Because this regression test has low power when meta-analyses include few studies (k < 10), it is sometimes advised to retain a significant level of *α* = 0.10 (see, e.g., the example in [104]), in which case 19 meta-analyses (14.6%) showed a significant small-study effect. We ran a robustness analysis including only meta-analyses with at least 10 primary studies and still found consistent evidence for a small-study effect (see Appendix A). We did not find consistent differences in the small-study effect between different types of studies. See Appendix A for details.

### 5.3. Other Bias-Related Patterns

We investigated the presence of several other bias-related patterns beside the small-study effect, following recent meta-science efforts in other fields (e.g., [31]): the decline effect, the US effect, and citation bias. A decline effect occurs when observed effect sizes decrease over time, potentially indicating overestimation in the first, more exploratory studies as compared to the more confirmatory follow-up studies ([99]; [106]). A US effect occurs when studies from the United States show stronger overestimation than studies from other countries, potentially caused by stronger publication pressure in the US ([23]; [29]; [109]). Finally, citation bias occurs when larger effects are cited more often than smaller effects, which could lend unjustified importance to studies with large effects ([19]; [24]). We explain these bias-related patterns in more detail below. Details about how we coded the relevant variables can be found in Appendix A.

We investigated the decline effect, US effect, and citation bias using the same general two-step meta-regressions we used to estimate the small-study effect. We summarized the specific meta-regression equations and the results of all bias analyses in Table 4. To correct for multiple comparisons, we applied a Bonferroni correction based on the number of predictors (four main patterns of bias, including the small-study effect) for our main meta-meta-regressions (see Table 4), resulting in a significance level of 0.0125.

**Decline Effect****:** A decline effect occurs when studies that are published earlier in a research line report larger effects, relative to later studies. One explanation of such a decline in effect sizes is that the first study/studies of a research line are smaller and more explorative in nature, which, combined with selective publishing/reporting practices, can lead to a higher risk of overestimated effects. Subsequent studies that are larger and more confirmative in nature will likely fail to find similar extreme effects, leading to a “decline” in effect size over time.[note 4] Previous research found evidence for decline effects in multiple scientific fields, including in intelligence research ([31]; [44]; [45]; [52]; [89], [90], [91]; [99]; [102]).

We estimated the decline effect following the two-step meta-regression approach, where we used the publication order as the predictor. Specifically, if a meta-analysis contained four primary studies published in 2011, 2012, 2012, and 2013, respectively, the publication order would be 1, 2, 2, 3, respectively. We found no overall evidence for a decline effect, *b^j^_PubOrder_* = 0.001, SE = 0.002, *Z* = 0.371, *p* = 0.711, 99% CI = [−0.003; 0.005], *I*^2^ = 53.7%, *var* = 0.00 (SE = 0.00). In six cases (4.6%) we found significant evidence for a decline effect against *α* = 0.05. We found no evidence that the decline effect was moderated by study type.

Our results on the decline effect are partly in line with recent results from [91] ([91]). Using meta-meta-regressions similar to the ones we presented above, they analyzed 29 meta-analyses (comprising 991 primary effects) published in the journal *Intelligence* and did not find significant evidence for an overall decline effect. [91] ([91]) performed additional analyses and concluded that a majority of individual meta-analyses did show evidence for a decline effect. We decided against interpreting patterns of bias in individual meta-analyses because individual meta-analyses often do not include sufficient studies to draw reliable conclusions about biases such as the decline effect.

In addition to the meta-meta-regressions, [91] ([91]) investigated whether the first study in a meta-analysis is overestimated compared to subsequent studies, also known as a “winner’s curse” ([47]). To test whether such a winner’s curse was present, we again used a meta-meta-regression approach in which we predicted effect size with a dummy-coded variable, indicating if a study in a meta-analysis was published first or not. We found no overall evidence for a winner’s curse in our larger set of meta-analyses, *b^j^_FirstPublished_* = 0.02, SE = 0.01, *Z* = 1.54, *p* = 0.123, 99% CI = [−0.015; 0.058], *I*^2^ = 29.8%, *var* = 0.00 (SE = 0.00).

We ran several additional analyses testing different time-related patterns in effect sizes, including the early-extremes effect (similar to the Proteus phenomenon, [47]; also see [91]) and several combinations of different covariates in our original analyses, but found no evidence for any systematic effects of time on effect size. We did find that across all intelligence meta-analyses, sample size seemed to increase with publication order. In other words, within a meta-analysis, studies that were published earlier had smaller samples than those published later. However, this effect was qualified by substantial heterogeneity, hence it may not generalize to all lines of intelligence research. See Appendix A for details on these additional analyses.

**US Effect:** Studies from the United States may have a higher probability of reporting overestimated effects ([29]; but see [80]). The suggested explanation for this “US effect” is that the publish-or-perish culture is stronger in the US than in other countries ([109]), which would make US researchers more inclined to take advantage of flexibility in data analysis ([98]) and select only (studies with) significant findings to submit for publication. To investigate whether such a US effect was present in the intelligence literature, we analyzed whether small-study effects were stronger in US studies than non-US studies (operationalized by an interaction between SE and country: US/non-US), potentially indicating stronger publication bias in the US ([23]; [80]).

We found a positive overall estimate of the interaction between SE and US on effect size, but this result was not statistically significant after we corrected for multiple testing (Bonferroni corrected α = 0.05/4 = 0.0125), *b^j^_US_**_*SE_* = 0.46, SE = 0.23, *Z* = 1.95, *p* = 0.051, 99% CI = [−0.15; 1.06], *I*^2^ = 4.2%, *var* = 0.20 (SE = 0.75). Of the 92 meta-analyses in which we could estimate a US effect, five (5.4%) showed significant evidence for a US effect (α = 0.05). In our analysis of the US effect, study type was not a significant moderator. Furthermore, we also did not find evidence for a US effect when we included only meta-analyses with at least five, instead of two, (non-) US studies. See Appendix A for details.

**Citation Bias:** Studies with larger effects may be more likely to be cited than studies with small, non-significant effects ([19]; [24]; [51]). Citation bias can cause effects to look more important or undisputed than they really are when taking into consideration all relevant evidence.

We did not find evidence for overall citation bias, *b^j^_CitPerYear_* = 0.008, SE = 0.005, Z = 1.693, *p* = 0.091, 99% CI = [−0.004; 0.020], *I*^2^ = 37.3%, *var* = 0.00 (SE = 0.00). We found significant evidence for citation bias in eight of the remaining 126 meta-analyses (6.3%; α = 0.05). We ran additional robustness analyses including several control variables, and consistently found no evidence for citation bias and no differences in citation bias between study types (see Appendix A for details).

**Robustness Checks:** For all four bias patterns (small-study effect, decline effect, US effect, and citation bias) we ran several robustness checks. We reran the meta-meta-regressions, including several potential covariates (e.g., SE, sample size, journal impact factor), but none of these robustness analyses showed different overall patterns than our main analyses. We also ran an “omnibus” meta-meta-regression, in which we estimated all bias patterns at once. We did not use this omnibus test as our main analytic strategy because adding more predictors to the model lowered the power and because we could only fit the full model for 82 meta-analyses (63%). We found patterns similar to the ones reported in Table 4. The main difference in this robustness check was that the estimate of the small-study effect was no longer significant (b^j^_SE_ = 0.57, 99% CI = [−0.07; 1.21]). The estimate of the coefficient itself, however, was similar to the one we found in the separate regression (b^j^_SE_ = 0.67, 99% CI = [0.35; 0.99]; see Table 4). Details of this “omnibus” meta-meta-regression can be found in Appendix A. The additional robustness checks are reported throughout the Appendix A under the headings of the relevant bias-related patterns.

## 6. Discussion

In this study, we analyzed 2442 effect sizes from 131 meta-analyses about intelligence published between 1984 and 2014, based on over 20 million participants. We found that the median effect size in this field was a Pearson’s correlation of 0.24 (unweighted mean *r* = 0.25). This effect was similar to the typical effect in psychology (r = 0.24; [8]; based on an average total sample size of 40 and d = 0.50). We found relevant differences between the subtypes of intelligence research. Specifically, we found that the types of studies that were least complex in terms of methodology were most often conducted and had the largest effect sizes; in correlational research and research about group differences we found median (unweighted) effect sizes of *r* = 0.26. In less prevalent—and arguably more complex types of research—the (unweighted) effect size was lower and decreased rapidly from *r* = 0.18 in experimental research, to *r* = 0.15 in toxicological studies, and *r* = 0.07 in behavior genetics. Given the typical effect sizes, the sample sizes for all study types were relatively small, with an overall median of 60.

Across all primary studies, we found an overall median power of 11.9% to detect a small effect, 54.5% for a medium effect, and 93.9% for a large effect (corresponding to a Pearson’s *r* of 0.1, 0.3, and 0.5 or a Cohen’s *d* of 0.2, 0.5, and 0.8, respectively). Again, we found relevant differences between study types; power was lowest in experimental research and highest in behavior genetics, although we noted that most of the observed effect sizes in behavior genetics research were much smaller than the standard small effect (Pearson’s *r* = 0.1 or Cohen’s *d* = 0.2) with which we calculated power. Overall, we concluded that all study types (except for behavior genetics) generally had samples too small to reliably detect a medium effect.

We found evidence for a relatively small but statistically significant small-study effect across the intelligence literature: On average, smaller studies seemed to yield higher effect sizes, although this effect was marked by substantial variance across meta-analyses. Not all meta-analyses showed a small-study effect and some meta-analyses showed a “large study effect,” i.e., larger sample sizes were associated with larger effect sizes. The presence of a small-study effect could be a sign of publication bias, especially given the overall low power we estimated. Note that even though a small-study effect is consistent with bias, it could also be explained by other factors, including true heterogeneity or chance (for an overview, see [103]). All five types of studies showed consistent evidence for a small-study effect, and we did not find evidence that the small-study effect was stronger in any of these types. We found no evidence that the small-study effect was stronger for US studies (the US effect) in intelligence research. This was in line with previous findings that the US effect does not seem robust against method of analysis ([31]; [80]). We also did not find evidence for citation bias. Finally, we did not find an overall decline effect, early-extremes effect, or other time-related biases.

Compared other fields, the potential problems in intelligence research seem less severe. First, the median power in intelligence research seems higher than the median power estimated in neuroscience (8–31%; [14]), psychology (between 12% and 44%; [101], [105]), behavioral ecology and animal research (13–16% for a small effect and 40–47% for a medium effect; [53]), economics (18%; [49]), and social-personality research (50% for *r* = 0.20; [33]). Second, we did not find trends in effect sizes over time, which might indicate that the field of intelligence research is less susceptible to time-lag biases such as the decline effect or the early-extreme effect ([106]). This was in line with the theory that such biases would affect mainly research fields in which results could be rapidly produced and published, which might not apply to the majority of studies about intelligence ([47]). Finally, citation bias seems to be a problem in medical research ([24]; [51]), and there is some evidence that it also affects social sciences in general ([31]), but in intelligence research, specifically, we found no evidence that larger effects were cited more often.

### 6.1. Limitations

#### 6.1.1. Sample

In our study, we were limited to meta-analyses that actually included the full data table, which was very often not the case (namely, in 81 meta-analyses). It is imaginable that the meta-analyses without data tables contained stronger and/or other patterns of bias. It could be the case that meta-analysts who go through the effort of presenting the full data in their papers are more rigorous in their work. This could then mean they may also have tried harder to find all primary studies (published and unpublished), which would have decreased overall bias in the meta-analysis.

Furthermore, not all studies in the intelligence literature end up being included in a meta-analysis. It is also important to note that the research question of a primary study does not necessarily have to be the same as the research question of the meta-analysis. If an included primary effect size does not correspond to the main hypothesis test from that study, there is less reason to expect bias in that effect size. Conversely, if an effect is the focus of a primary study, there may be more pressure to report significant and/or inflated effect sizes. It is possible that the relatively weak bias patterns we found in this study can be explained by differences in focus in primary studies and meta-analyses.

Our sample of meta-analyses included a wide range of years of publication (1984–2014, with primary studies covering almost 100 years), but did not include meta-analyses published later than 2014. Since 2012, there has been increased attention paid to problems concerning overestimated effect sizes and low power in psychology ([85]; [86]). As a result, biases may be weaker in more recent articles and in more recent meta-analyses, as well. However, studies in any meta-analysis normally cover many years, so even if there have been improvements in recent years, it will likely take a while before these improvements are visible in meta-analyses.

We did not have access to some meta-analyses that did match our search criteria because they were behind a paywall or not written in English. It is imaginable that there are systematic differences between these meta-analyses and the ones we did include. However, we suspect that many of these meta-analyses would have been excluded in later steps, anyway, because of overlap with other meta-analyses or missing data tables, which means that only a handful of meta-analyses would have been eligible for inclusion. Furthermore, a post-hoc analysis of this set of excluded records showed wide variation in publication year, journal impact factor, research question, and study type, so we see no immediate reason to think that this excluded sample would systematically differ from our included sample. We therefore do not expect that the exclusion of this set of meta-analyses affected our overall conclusions.

#### 6.1.2. Analyses

It is important to note that we estimated many meta-regressions for meta-analyses with only few primary studies in them. This means that the individual bias estimates for each of the meta-analyses may be unreliable ([96]). We therefore did not draw conclusions about the individual meta-analyses but summarized the obtained meta-regression coefficients in a single estimate for each of the bias patterns we investigated. It is also possible that these meta-meta-regressions are underpowered (see [30]), so any significance tests on our data need to be interpreted with care. In future research, it would be valuable to garner an even larger sample of meta-analyses, conduct formal power analyses for the (preferably preregistered) meta-meta-regressions, and consider other options for modeling the different types of bias.

When interpreting our current results, it is also important to take into account that these are patterns of potential bias that are aggregated over meta-analyses. Even though we found evidence for an overall small-study effect, this does not mean that each meta-analysis in intelligence research showed this problem. Conversely, even though we did not find evidence for an overall decline effect, US effect, or citation bias, this does not mean that these problems never occur in intelligence research. Furthermore, there are other types of scientific biases that we did not investigate here. For instance, previous studies showed evidence for a “grey literature bias” ([22]; [31]; [38]; [69]; [99]). Here, unpublished literature such as Ph.D. theses or conference proceedings typically report smaller effects than research published in peer-reviewed journals, which could be a possible indicator for publication bias. Another type of bias we did not investigate is “industry bias,” where sponsorship from a company may be related the size and direction of published effects ([31]; [62]). These might be interesting patterns to investigate in future research.

### 6.2. Conclusion and Recommendations

Based on our findings, we conclude that intelligence research from 1915 to 2013 shows signs that publication bias may have caused overestimated effects. Specifically, we found that in these years power was often too low to reliably detect small or medium effects and, in general, smaller studies yielded larger effects. This is in line with the notion that publication bias and perhaps also researcher degrees of freedom in the analysis of data and reporting of results may have led to overestimated effects. Even though there might be several alternative explanations for these results, we argue that it is safe to assume that intelligence research has not been immune to the problems of robustness in psychology, although the problems in intelligence seemed to be less severe as compared to other fields.

There are several strategies to improve the reliability of primary studies and meta-analyses ([3]; [13]; [75]). Some potential improvements entail top-down changes: We might need systematic changes in peer review, in allocation of funding, and in university policies. However, there are also multiple things researchers themselves can do, bottom-up ([84]). One bottom-up strategy to improve reliability of published findings is to increase power ([14]; [57]; [81]). Strategies to increase power include increasing sample sizes or increasing reliability of measurements. When researchers run a power analysis to determine the required sample size, researchers should think carefully about the effect size they expect to find. The standard categories of Cohen (i.e., correlations of 0.1, 0.3, and 0.5 corresponding to small, medium, and large effects, respectively) might not be realistic in every line of research (see, e.g., [37] for empirical suggestions for small, medium, and large effects in individual differences research). Furthermore, researchers need to take into account that published effect sizes from previous research are probably overestimated. Ways to deal with this are correcting the observed effect sizes for publication bias ([2]; [108]; [112]), calculating lower bound power ([87]), or basing a power analysis on the smallest effect size of interest ([26]). Another improvement would be to avoid publication bias and opportunistic use of flexibility in data analysis by preregistering study plans ([17]; [114]). A final recommendation is to increase transparency by sharing materials, data, and analysis scripts to facilitate replication and reanalysis ([79]; [83]; [116]). This list of recommendations is far from exhaustive: over the years, many solutions have been suggested, targeting all steps in the empirical cycle. We feel optimistic that many of these strategies will help improve our research practices and scientific output.

## Figures and Tables

**Figure 1 jintelligence-08-00036-f001:**
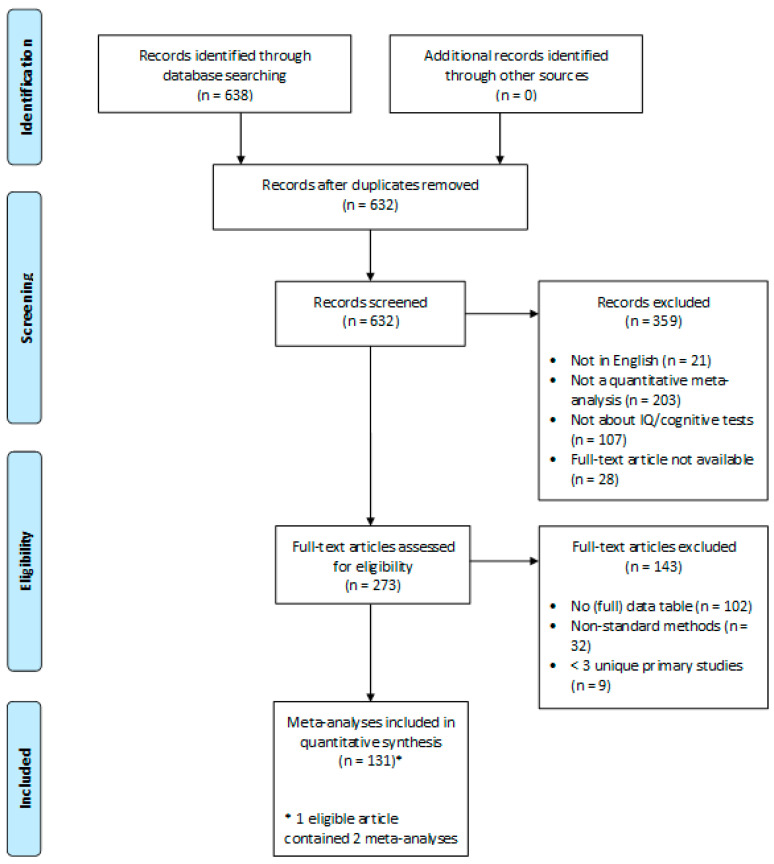
PRISMA Flow Diagram of the number of records identified, included and excluded, and the reasons for exclusions ([74]).

**Figure 2 jintelligence-08-00036-f002:**
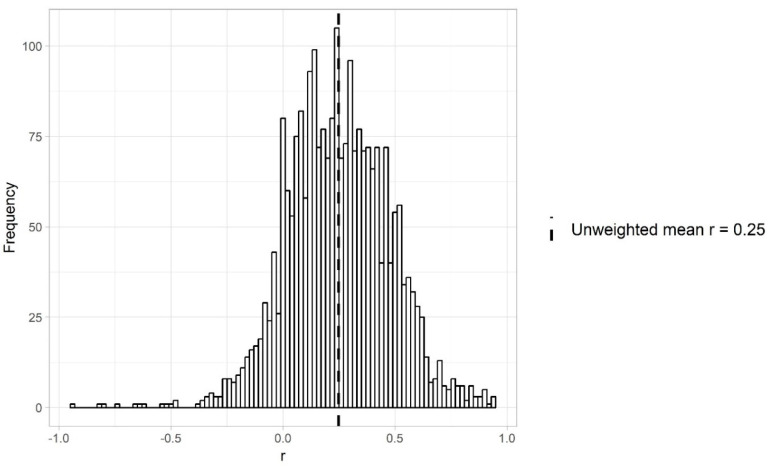
Histogram of the effect sizes of 2442 primary studies about intelligence. All effect sizes were converted from Fisher’s *Z* to Pearson’s correlation to facilitate interpretation.

**Figure 3 jintelligence-08-00036-f003:**
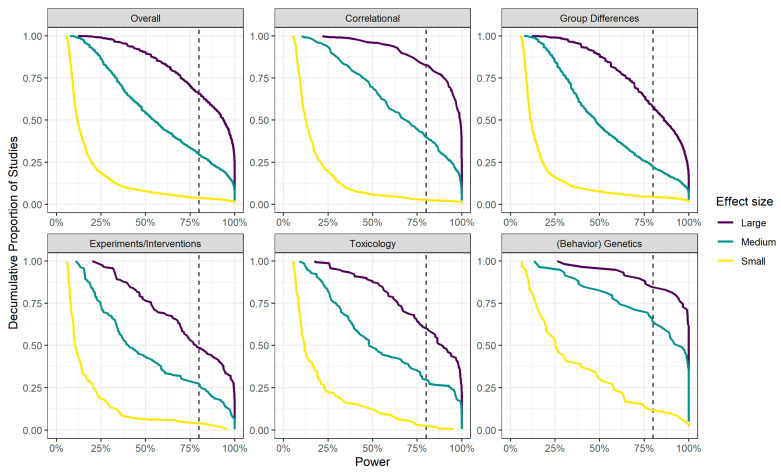
Decumulative proportion of primary studies that had at least a certain power to detect a large, medium, or small effect, split up per study type and overall. The vertical dotted line indicates the nominal power of 80%. Small, medium, and large effects correspond to a Pearson’s *r* of 0.1, 0.3, and 0.5 or a Cohen’s *d* of 0.2, 0.5, and 0.8, respectively.

**Table 1 jintelligence-08-00036-t001:** The number of included meta-analyses and primary studies per type.

Type of Research	Explanation	# Meta-Analyses	# Unique Primary Studies
1. Correlational studies	(a) Selected IQ test is correlated with other, continuous measurement of psychological construct; (b) test–retest correlation.	31	781
2. Group differences (clinical and non-clinical)	Correlation IQ test and categorical, demographical variables or clinical diagnoses (e.g., male/female, schizophrenia yes/no).	59	1249
3. Experiments and interventions	Studies in which participants are randomly assigned to conditions to see if the intervention affects IQ.	20	188
4. Toxicology	Studies in which IQ is correlated to exposure to possibly harmful substances.	16	169
5. Behavior genetics	Genetic analyses and twin designs.	5	59

Note. We categorized the meta-analyses in five different types, reflecting substantive differences in research questions and methodology, while retaining a sufficient number of meta-analyses within each type.

**Table 2 jintelligence-08-00036-t002:** Descriptive statistics of the primary studies split up into five types of studies and in total.

	# Meta-Analyses	# Unique Primary Studies	Total N	Median N	Range N	Median Unweighted Pearson’s *r*	Median Meta-Analytic Effect (*r*)
1. Predictive validity and correlational studies	31	779	367,643	65	[7; 116,053]	0.26	0.24
2. Group differences (clinical and non-clinical)	59	1247	19,757,277	59	[6; 1530,128]	0.26	0.19
3. Experiments and interventions	20	188	24,371	49	[10; 1358]	0.18	0.17
4. Toxicology	16	169	25,720 ^a^	60	[6; 1333]	0.15	0.19
5. (Behavior) genetics	5	59	30,545	169	[12; 8707]	0.07	0.08
**Total**	**131**	**2442**	**20,205,556**	**60**	**[6; 1530,128]**	**0.24**	**0.18**

Note: “N” indicates number of participants in a primary study. We calculated the meta-analytic effects per subtype by taking the median of the random effects meta-analyses estimates. ^a^ One of the meta-analyses reported two studies with non-integer total sample sizes. It seems that the authors wanted to correct their sample sizes to ensure they did not count the same observations twice. Here, we rounded the total sample size.

**Table 3 jintelligence-08-00036-t003:** Median power of primary studies in intelligence research to detect a small, medium, and large effect, split up per study type and overall.

Study Type	Median Power to Detect a … Effect *		
	Small	Medium	Large
1. Predictive validity and correlational studies	12.5%	68.2%	99.1%
2. Group differences (clinical and non-clinical)	11.4%	47.7%	86.1%
3. Experiments and interventions	10.5%	39.9%	77.9%
4. Toxicology	11.9%	47.9%	88.2%
5. (Behavior) genetics	25.1%	92.0%	100.0%
**Total**	**11.9%**	**54.5%**	**93.3%**

* Small, medium, and large effects correspond to a Pearson’s *r* of 0.1, 0.3, and 0.5 or a Cohen’s *d* of 0.2, 0.5, and 0.8, respectively.

**Table 4 jintelligence-08-00036-t004:** Overview of the meta-meta-regressions we estimated in this paper to investigate different predictors for effect size that could potentially indicate bias. We estimated these bias-related patterns in five separate analyses.

Type of Bias	“Predictor” in $Fisher^{\prime }s\;Z_{ij}=a^{j}+b^{j}Predictor_{ij}+\varepsilon_{ij}$	Included Number of Meta-Analyses (m) and Number of Primary Effect Sizes (k)	Estimate of the Mean Parameter Across Meta-Analyses [99% CI]	Variance in the Meta-Regression Slopes (SE)
1. Small-study effect	Standard error of primary study’s effect size (SE)	m = 130; k = 2432	0.67 [0.35; 0.99]	*var* = 0.73 (0.25)
2. Decline effect	Order of publication	m = 131; k = 2442	0.001 [−0.003; 0.005]	*var* = 0.00 (0.00)
3. US effect	US × SE	m = 92; k = 2114	0.46 [−0.15; 1.06]	*var* = 0.20 (0.75)
4. Citation bias	Citations per year (log transformed)	m = 126; k = 2405	0.008 [−0.004; 0.020]	*var* = 0.00 (0.00)

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
