# Peer review of "Effect Sizes, Power, and Biases in Intelligence Research: A Meta-Meta-Analysis"

_jintelligence, 2020, doi:10.3390/jintelligence8040036_

Round 1

Reviewer 1 Report

This paper 2,442 effect sizes from 131 meta-analyses in intelligence research, published from 1984 to 2014, to estimate the average effect size, median power, and evidence for bias. The paper is clear, transparent, and well-written, and I can see a revised version of it making a nice contribution. I have three broad, overarching concerns followed by a number of more focused comments.

First, I am concerned about the contribution over Pietschnig et al (2019). The present paper contains more meta-analyses but they none later than 2014 while Pietschnig et al includes meta-analyses up to 2018. Further, it seems the present results are pretty consistent with those in the prior paper (e.g., see page 10 of the present paper).

Second, the authors claim “evidence for bias” based on their small-study analysis. However, if anything, the authors’ analysis shows such bias is extremely weak even negligible. Specifically, the authors write:
“[T]he estimated overall small study effect of bSE = 0.68 across meta-analyses means that if the total sample size of a study increases from 50 to 100 observations, the estimated Fisher’s Z decreases from .24 to .21 (corresponding to r = .24 and r = .21, respectively).”
In other words, if the sample size _doubles_ from the median in intelligence research*, r decreases only _trivially_ (of course this is an estimate from a model which has uncertainty attached to it, so the decrease could be even smaller). My conclusion is thus the authors show bias is not important in this line of research, and therefore the authors’ claim should be reversed.
* The median sample size in intelligence research is actually 60 as per Table 1 but the 50 in the sentence is close enough.

Third, the analyses around retrospective meta-analytic power should be cut. The problem is that retrospective meta-analytic power is an extremely noisy quantity. To see this, imagine one has 19 effect sizes in a meta-analysis and each has exactly the same power of 51.7% (i.e., the authors’ 2442 effect sizes divided by 131 meta-analyses gives about 19 effect sizes per meta-analysis and they report median power of 51.7%). The estimate of median power estimated from this setup can range from 17% to 84% and 95% of the time falls between 34% and 69%. In other words, the typical meta-analysis in the authors’ sample produces an extremely noisy estimate of median power—and this even when there is no heterogeneity, no publication bias, no flexibility in data analysis, etc. Of course, adding in the effect of all these factors which are almost certain to be a reality in the authors’ data will make these estimates of power even noisier. Further, given that three of the authors’ five “types” have more like 10 studies per meta-analysis, the estimates are also noisier there (95% of them ranging from 11% to 89%).

Now, I realize the authors estimate the power across many meta-analyses not just one. However, when one also accounts for the additional factors above (heterogeneity, publication bias, flexibility in data analysis, etc.) and the fact that the meta-analyses themselves are almost certain to have different median powers, it may not make the resulting estimate any less noisy. Therefore, we cannot at present trust it without a wholesale investigation of the estimates.

More focused comments:

1. Page 2: Why stop data collection at 2014? Why not continue it?

2. Page 2: This statement is much too strong and needs nuance:
“Based on these findings, we conclude that it is unlikely that these excluded articles are systematically different from the included ones, so we do not suspect that these articles would affect our conclusions substantially.”

3. Page 2: This seems unjustified:
“Finally, we excluded 28 records that were only available in print format, or put behind paywalls that our library had no subscription to.”
Can the authors get colleagues at other institutions to help you?

4. Page 2-3: The authors write:
“Because of the positive manifold between cognitive tests and fairly high correlations between broad cognitive factors [31], this strategy ensures inclusion of measures bearing on intelligence, while also creating less heterogeneity within meta-analyses that would have been present had we included entirely different (types of) cognitive tests.”
This seems correct, but it should be noted that effect sizes based on subtests rather than the full IQ test will be noisier; ideally this could be corrected for in the analysis.

5. Page 3: This seems unjustified:
“We excluded … 32 meta-analyses that used non-standard meta-analytic methods (e.g., multi-level models based on individual level data or unweighted analyses).”
In fact, meta-analysis based on individual level data using multi-level models is considered by many to be a gold standard in meta-analysis.

6. Page 3: This also seems unjustified:
“Finally, we excluded 9 meta-analyses that contained fewer than three unique primary studies.”

7. Page 3-4: This also seemed unjustified (and ad hoc):
“If a study appeared in more than one meta-analysis, we removed it from the meta-analysis with the most primary studies.”
This means the meta-analyses the authors are analyzing are not the meta-analyses conducted by the original meta-analysis authors. Also, depending on how many studies were removed in this way, the sequential approach used could end up leading to different results. Ideally the authors should account for the dependence induced by studies appearing in multiple meta-analyses using multilevel or related techniques. At the very least, we need a lot more information about this issue.

8. Related to the above, how about studies that reported multiple effect sizes.? Again, ideally the authors should account for the dependence induced by this. At the very least, we need a lot more information about this issue.

9. It would be nice to have the histogram of effect size estimates and sample sizes for the data as a whole and for each of the five types. The effect sizes for the data as a whole are currently in Figure 2, but the others would make for a welcome addition.

10. Page 7: I did not follow the sentence:
“The fact that certain research types occur more often in the literature (at least as included in meta-analyses), might also explain why there are more meta-analyses in these fields.”
It seemed almost tautologous to me so I think I must not be understanding it.

11. Page 9: Given that the term post hoc power is also used for observed power (i.e., retrospective power) as per footnote 1, I think the authors should use a different term. In my opinion, the use of post hoc echoing that of Erdfelder is the more idiosyncratic usage and post hoc typically means observed. Of course, conceptually they are the same, except one is post hoc observed power based on one study and the other is based on many studies.

12. Page 10: The authors write
“We then calculated the power of each primary study to detect the corresponding meta-analytic effect.”
It is well known that heterogeneity causes the power at the meta-analytic effect (i.e., what the authors use) to be inflated above the actual expected value of power. See
Kenny DA, Judd CM. The unappreciated heterogeneity of effect sizes: Implications for power, precision, planning of research, and replication. Psychol Methods. 2019 Oct;24(5):578-589. doi: 10.1037/met0000209. Epub 2019 Feb 11.
which reviews some literature on this topic.

13. Page 10: I believe the wide error bars in Figure 3 are related to my third major comment.

14. Page 12-13: There is no reason to use a two-step estimation approach when the authors could estimate this model singly using a multilevel model.

15. Page 15 / Table 3: Why do the heterogeneity estimates vary so dramatically across the four rows of this table given the massive overlap in data? This seems highly peculiar and would only make sense if the moderators were strong predictors, which the effect estimates show they are not. Indeed, it is extremely peculiar that heterogeneity is reduced to zero for two statistically insignificant moderators.

16. Page 16-17: More generally, I am not sure the four moderator analyses (or even the small study analysis) are necessary. They do not seem to add much to the main story. However, if they are going to be included, the “omnibus” analysis approach the authors discuss seems far more appropriate than the piecemeal approach they focus on.

17. Page 16: What are the units for the decline effect? Per year?

18. Page 16: The authors write:
“We found a positive overall estimate of the interaction between ES and US on effect size.”
Do they mean SE rather than ES? I think so based on what they write two lines above but am not sure.

19. Page 17: The last paragraph about the goal of the manuscript seems entirely unnecessary. The writing is really good and this goal was clear throughout the manuscript.

20. Page 19: The following comment
“Related, the current study was not preregistered and should therefore be considered explorative in nature.”
seems very strange. While there are very narrow circumstances where preregistration is both possible and desirable, this study is certainly not one of them.

21. Page 19: The authors write:
“Strategies to increase power include increasing sample sizes or increasing reliability of measurements.”
Given that we are talking about IQ which is studied using well-calibrated and validated materials that are common across studies, I do not think the suggestion about increasing reliability applies here as it does in, say, social psychology. The same holds a few sentences later when the authors write
“A final recommendation is to increase transparency by sharing materials, data, and analysis scripts, to facilitate replication and reanalysis”
at least as it applies to materials and data. I think these should be removed.

Author Response

Dear Reviewer,

Thank you for your thoughtful comments. Please find our reply attached.

Best wishes,

Michèle Nuijten

Reviewer 2 Report

This is a well written manuscript that informatively addresses a relevant issue. It therefore has the potential to make a valuable contribution to improving our practice of conducting empirical psychological research. The manuscript reports in a sufficiently transparent manner with adequate levels of detail.  The only concern I am having sits with the chosen typology of research. In its current form this categorisation confounds substantive-focussed (sub-discipline, or research field) with methodological (design and methods) focussed perspectives. A framework such as this might nurture the impression of dealing with mutual exclusive categories, which would erroneously suggest that studies in behaviour genetics do not employ correlational designs, or that quasi-experimental designs are not to be found in toxicological research. I therefore strongly suggest a categorisation of studies that is based on the research design that was employed. This should result in a differentiation between correlational, quasi-experimental, and experimental studies.  I believe that to be much more meaningful and informative. The question, for example, whether research employing (quasi-)experimental designs in toxicology yield different ES patterns than studies trialing interventions would be a secondary perspective that could still be addressed. As another example for the benefit of employing a design-based categorisation that is not confounded by considerations of substantive groundings of the research in question: It would enable contrasting ES patterns of research that tends to identify more with the null hypothesis (e.g., testing for harmlessness) on one the hand with studies that seek to establish evidence for rejecting the null hypothesis (e.g., testing for effectiveness of interventions) on the other.   In my opinion, the proposed reconsideration of the categorisation schema would make a very good manuscript even stronger.

Author Response

(The authors gave the same response as above.)

Round 2

Reviewer 1 Report

This paper 2,442 effect sizes from 131 meta-analyses in intelligence research, published from 1984 to 2014, to estimate the average effect size, median power, and evidence for bias. The paper is clear, transparent, and well-written, and I can see a revised version of it making a nice contribution. I have three broad, overarching concerns followed by a number of more focused comments.

In this review, I revisit my three overarching comments from the first round as well as your response to them. I then make a number of more focused comments.

My first comment was around contribution. I appreciate the authors’ response. It satisfies me if it satisfies the editor and other reviewers.

My second comment was that the authors claim “small-study bias” but their analysis shows such bias is extremely weak even negligible. I noted that a doubling in sample size from 50 (roughly the median in this research) to 100 is associated with a trive decrease r from 0.24 to 0.21. The authors responded by inserting the following text:

“Concretely, the estimated overall small study effect of bSE = 0.67 across meta-analyses means that two otherwise identical studies with sample sizes of 25 and 100 observations would estimate Pearson’s r equal to .3 and .2, respectively. Such differences in effect size estimates could influence substantive interpretations and practical implications of results in single studies and meta-analyses.”

I strongly disagree. Let’s dive into the example you give in the text above. First, recall, the standard error of a correlation is roughly 1/sqrt(n). Given this, your hypothetical studies of [n=25, r=0.3] and [n=100, r=0.2] would be accompanied by SEs of 0.2 and 0.1 respectively. Consequently, the two studies are entirely consistent (e.g., the p-value on the null of zero difference between the two is 0.37). Thus, in no way should this “influence substantive interpretations and practical implications of results” as you claim. See also 7 below.

You have found a ‘statistically significant’* but practically irrelevant small study effect. That’s fine! The results are the results! Actually, this is really good news for the field of intelligence research!

[* Note: even "statistical significance" is questionable given the results of the preferable omnibus model reported on page 17.]

My third comment was around the retrospective power calculations. I commend the authors decision to remove these and replace them with the small / medium / large effect size power calculations. I have one very focused comment about a single sentence in the single paragraph around the retrospective power calculations that remains. You write:

“It is important to note that these power estimates are likely noisy due to heterogeneity in the true effect sizes [69,70].”

This is not quite right. To be precise, the power estimates are noisy due to the fact that they are based on a relatively small number of studies (i.e., the point of your reference 69, McShane, B.B.; Böckenholt, U.; Hansen, K.T. Average power: A Cautionary Note). And they are biased due to heterogeneity (i.e., the point of your reference 70, Kenny, D.A.; Judd, C.M. The unappreciated heterogeneity of effect sizes as well as its predecessor paper McShane, B.B. and Böckenholt, U. You Cannot Step into the Same River Twice).

More focused comments follow:

  1. Throughout the paper, you frequently use the word “power” (and related words like underpowered). In many of these contexts, you specifically mean power but in many other context you mean something more like “small sample size” or “imprecise.” I’d case that you go through the manuscript for all uses of the word power and cognates and carefully consider whether you mean power in the strict sense or whether you mean something more general.
  1. Would it be possible to report how much overlap there is between your 2,442 effect sizes and the 1,391 of Pietschnig, et al.
  1. I still am a bit unclear about your removal of studies that appeared in more than one meta-analysis. I realize your principal results are at the study-level so this is simply a deduplication for that purpose. However, there are a few questions I have that relate to some further analyses:

3a. Your removal procedure (removing it from the biggest) introduces a sequential dependence. For example, had I started with Study X as the first multiple-instance study, I would have removed it from Meta-analysis 12, say. However, had I started with Study Y and Study X came much later, Meta-analysis 12 could have already had many studies removed from it and so maybe I would now remove it from Meta-analysis 22, say. This seems unsatisfactory.

3b. Do any of the multiple-instance studies appear in meta-analyses in across more than one of your 5 types? If so, I would argue that they should NOT be removed for analyses such as that in Table 1 and 2. I think it would be perfectly fair, in fact correct, to count them once in each types.

3c. The removal process has a weird effect on your meta-regression results. Consider your Equation 1. Because you might remove studies from meta-analysis j, then the (a^j, b^j) you estimate are not what would be estimated from the studies in meta-analysis j or by the authors of meta-analysis j. This is obviously far from ideal. Now, I know it introduces dependence in your second stage analysis if a given study from meta-analysis j is also included in meta-analysis k. However, if only a few studies show up multiple times, ignoring this dependence this might be the lesser evil because it will have only a small effect.

  1. Figure 3 is beautiful!
  1. Obviously a one-step multilevel model is superior to your two-step approach, but I understand your reasons for employing the two-step here. However, I wonder if in the second stage, rather than modeling only the b^j in a univariate meta-analysis if you would model the (a^j, b^j) pairs in a bivariate one. I think this could add some precision to your estimates depending on how the data patterns and the modeling approach employed.
  1. Also, you say you use random effects for both stages, but what in particular is treated as random in each stage? Can you please be a bit more specific?
  1. Assuming I am interpreting things correctly, you find an average small study effect of 0.67 with heterogeneity across meta-analyses of sqrt(0.73) = 0.85. This is tremendous variation, especially compared to the average of 0.67. I think the real story here is not in the average that is your focus but on the variation across meta-analyses. In fact, your results suggest ~20% of meta-analyses have a “large study effect” rather than a small one!
  1. Page 17: You write:

“We did not find evidence for overall citation bias, bjCitPerYear = 0.008, SE = 0.005, Z = 1.693, p = .091, 99% CI = [-0.004; 0.020], I2 = 37.3%, var = 0.00 (SE = 0.00)”

You cannot take failure to reject the null as proof or confirmation of the null. This is a well-known fallacy.

Author Response

We thank the reviewer for the thoughtful and detailed comments. We have addressed them all and include our point-by-point response in the attached Word file (in order to retain the text formatting).

Round 3

Reviewer 1 Report

This paper 2,442 effect sizes from 131 meta-analyses in intelligence research, published from 1984 to 2014, to estimate the average effect size, median power, and evidence for bias. The authors have been quite responsive to my comments and I thank them for that. I have two very small comments. They both pertain to the following passage on page 29:

“We found evidence for a relatively small but statistically significant small study effect across the intelligence literature: smaller studies seemed to yield higher effect sizes, although individual meta-analyses varied substantially in the extent to which they displayed such a small study effect.”

1. The current wording “although individual meta-analyses varied substantially in the extent to which they displayed such a small study effect” implies all meta-analyses do show such an effect, but simply that it is stronger versus weaker in some versus others. But, the variance of the meta-regression slopes in Table 4 show this is not the case: some meta-analyses will have a near zero effect and some a “large study effect.” The language should be revised so that is does not imply all meta-analyses show a small study effect.

2. Other places in the manuscript that discuss this result (e.g., the abstract and introduction) should be revised in a similar manner as the above to note the variation across meta-analyses.

Author Response

Thank you for your comments. Please find our reply attached.
